# Foci-Xpress: Automated and Fast Nuclear Foci Counting Tool

**DOI:** 10.3390/ijms241914465

**Published:** 2023-09-23

**Authors:** Jae-I Moon, Woo-Jin Kim, Ki-Tae Kim, Hyun-Jung Kim, Hye-Rim Shin, Heein Yoon, Seung Gwa Park, Min-Sang Park, Young-Dan Cho, Pil-Jong Kim, Hyun-Mo Ryoo

**Affiliations:** 1Department of Molecular Genetics and Dental Pharmacology, School of Dentistry and Dental Research Institute, Dental Multi-Omics Center, Seoul National University, Seoul 08826, Republic of Korea; moonjaei@snu.ac.kr (J.-I.M.); carpediemwj@snu.ac.kr (W.-J.K.); kitae@snu.ac.kr (K.-T.K.); hkim7@snu.ac.kr (H.-J.K.); dnfckdwjd@snu.ac.kr (H.-R.S.); heein0116@snu.ac.kr (H.Y.); madirene@snu.ac.kr (S.G.P.); titanium1253@snu.ac.kr (M.-S.P.); 2Epigenetic Regulation of Aged Skeleto-Muscular System Laboratory, School of Dentistry and Dental Research Institute, Seoul National University, Seoul 08826, Republic of Korea; 3Department of Periodontology, School of Dentistry and Dental Research Institute, Seoul National University, Seoul 03080, Republic of Korea; cacodm1@snu.ac.kr; 4Department of Biomedical Knowledge Engineering Laboratory, School of Dentistry and Dental Research Institute, Seoul National University, Seoul 08826, Republic of Korea

**Keywords:** foci quantitative analysis, image processing, nuclear foci, automated screening method, DNA damage

## Abstract

In the nucleus, distinct, discrete spots or regions called “foci” have been identified, each harboring a specific molecular function. Accurate and efficient quantification of these foci is essential for understanding cellular dynamics and signaling pathways. In this study, we present an innovative automated image analysis method designed to precisely quantify subcellular foci within the cell nucleus. Manual foci counting methods can be tedious and time-consuming. To address these challenges, we developed an open-source software that automatically counts the number of foci from the indicated image files. We compared the foci counting efficiency, velocity, accuracy, and convenience of Foci-Xpress with those of other conventional methods in foci-induced models. We can adjust the brightness of foci to establish a threshold. The Foci-Xpress method was significantly faster than other conventional methods. Its accuracy was similar to that of conventional methods. The most significant strength of Foci-Xpress is automation, which eliminates the need for analyzing equipment while counting. This enhanced throughput facilitates comprehensive statistical analyses and supports robust conclusions from experiments. Furthermore, automation completely rules out biases caused by researchers, such as manual errors or daily variations. Thus, Foci-Xpress is a convincing, convenient, and easily accessible focus-counting tool for cell biologists.

## 1. Introduction

Nuclear foci are localized concentrations of proteins or other cellular components that form within the cell nucleus. The formation and disassembly of nuclear foci are crucial for regulating cellular processes, influencing cell function and survival. The study of these foci helps biologists understand cellular processes and elucidate the mechanisms involved in the onset and progression of diseases [1,2,3].

Manual counting is the most popular quantification method for nuclear foci. However, this method is time-consuming, and there is a possibility of conditioning owing to researcher biases. To reduce these biases, researchers may select other analytical platforms. Practically, one of the most widely used image analysis methods is unbiased direct counting by biologists [4]. This method achieves accurate and stable results. However, it is time-consuming and unsuitable for analyzing large-scale samples. It can also blur the researcher’s criteria. To overcome this problem, numerous researchers have attempted to quantify these foci accurately and promptly [5,6,7,8,9]. Image analysis software includes ImageJ (Fiji) and CellProfiler [10,11,12,13]. ImageJ is an open-source image processing and analysis program, and Fiji is one of its distributions. They provide plugins for analyzing nuclear foci, such as the ‘Find maxima…’ feature that can identify and analyze foci in images. CellProfiler is an open-source image analysis software written in MATLAB that offers various modules for measuring the size, shape, and density of cells and nuclear foci [11,12,13]. However, these software programs have lower foci recognition accuracy compared to manual counting, and the time-saving effect is not significant, as individual analyses are still required for each image.

Nuclear foci provide localized regions where multiple proteins and cellular components can interact with specific cellular responses and functions. For validation purposes, we employed two well-established model systems to assess the efficacy of our analytical tools. These models involved the induction of subcellular foci, namely deoxyribonucleic acid (DNA) damage-induced phosphorylated histone H2AX (γH2AX) foci and telomere dysfunction-induced foci (TIF).

In DNA damage and repair, nuclear foci serve as areas where proteins involved in repair concentrate during DNA damage, with and specific foci such as γH2AX. Sites of double-strand breaks (DSBs) are labeled with γH2AX, which is a representative biomarker of DNA DSB [14,15]. γH2AX is a form of the histone H2A variable (H2AX) protein. Phosphoinositide 3-kinase-like kinase (PIKK) family proteins, such as ataxia telangiectasia mutated (ATM), ataxia telangiectasia, and rad3-related (ATR) phosphorylate, recognize the S139 sequence of the H2AX protein [15,16].

In another model of nuclear foci formation, there is a phenomenon in which damaged response proteins gather at the telomere region, commonly induced by telomere dysfunction. This phenomenon is known as TIF [17,18,19,20,21]. TIF analysis involves the use of the fluorescence in situ hybridization (FISH) method to detect the localization of the damage marker, such as γH2AX, and the telomere sequences [22]. This approach not only facilitates the understanding of a biological mechanism in which cellular responses are elicited due to disturbances in telomere function, but also provides valuable insights into the dynamic interplay between damage-induced foci, such as γH2AX, and their colocalization with telomere regions. The observation of such colocalization events can offer diverse inspirations, shedding light on the intricate relationships between damage signaling and telomere maintenance mechanisms.

In this paper, we developed Foci-Xpress, an open-source tool that provides automated counting and analysis of nuclear foci. Foci-Xpress enables the processing of multiple foci images in a single run through automation of the counting process and allows for the analysis of nuclear size, foci intensity, and number with minimal parameter settings. For validation, we used a set of foci images (~100 images) labeled with γH2AX and 4′,6-diamidino-2-phenylindole (DAPI) in a cellular model subjected to oxidative stress. The results demonstrated that Foci-Xpress exhibited an accuracy similar to that of manual counting, and the analysis speed was significantly higher than that of manual counting or Fiji. Foci-Xpress is easily accessible for biologists and can be utilized for the accurate and rapid analysis of large image sets in various biological studies and the discovery of new drug candidates.

## 2. Results

### 2.1. Foci-Xpress Manual

The script is designed to be compatible with the ImageJ program and is obtained in three major steps (Figure 1A, left panel). The output derived from Foci-Xpress for images taken using confocal microscopy is formed independently of the folder. The regions of interest (ROI) are stored as other images, setting the image process and foci capture process using DAPI in the folder (Figure 1A, middle panel). Owing to the system, the results for the files set to input are generated in a ‘CSV’ format (Figure 1A, right panel), displaying the number of nuclei (number_total_nuclei). The value for “number_selected_nucleus” in the CSV file is represented by the degree of damaged cells, and these values are shown as values between 0 (no damaged cells) and 1 (all cells are damaged cells). In total, four CSV files are generated, including information for each wavelength and for the minimum and maximum values of colocalization between wavelengths. The CSV file index lists the names of the CZI files taken using confocal microscopy in the first column, followed by columns for “average_intensity”, “number_selected_nucleus”, “number_total_nuclei”, and “ratio_selected_nucleus”. “Average_intensity” represents the average value of the total intensity present in the nuclei. “Number_selected_nucleus” indicates the nucleus selected by the parameter set before running the process, whereas ”number_total_nuclei” is the total number of recognized nuclei in the image taken using confocal microscopy. “Ratio_selected_nucleus” is the ratio of “number_selected_nuclei” to the “number_total_nuclei.” Additionally, there is a folder for each image name, and inside each folder, the analysis results for each image with a detected nucleus are stored in a CSV file format. These CSV files contain the minimum, maximum, and mean values of the intensity of each focus in the nucleus, separated by wavelength.

The Foci-Xpress interface is also shown in Figure 1B. To proceed with the tool, click on “Run” from the menu located at the top of the main interface. Subsequently, upon providing the file path of the image and an alternate file path for receiving the analytical outcomes, one can obtain the results of the analysis. The image file must contain a CZI extension. The status interface for entering parameters appears when pressing "Run” on Foci-Xpress. The tool uses the minimum nucleus size (μm^2^) for foci counting (nucleus size), focus intensity (mark brightness), and the minimum number of Foci above mark brightness (mark count) as parameters.

### 2.2. Foci-Xpress Can Measure Damaged Cell Counting Accurately and Quickly

As described in the Materials and Methods section, the cells imaged using confocal microscopy can only be set to an ROI of over 100 μm^2^ along the nucleus’s boundary using the Foci-Xpress script dragged into ImageJ (Figure 2A, left panel). The ROI was layered on the image representing γH2AX foci (Figure 2A, lower left panel) to convert the foci in the ROI to black and white. These black and white images can have a brightness intensity set from a 16−bit image within the range of 0−65535 and a count brightness above a preset value known as “mark brightness” (Figure 2A, lower right panel). When the number of specified foci exceeds the number set by the mark count, it is recognized as a damaged cell, and the cells are counted in the macro system (Figure 2A, right panel). A total of 50 μM of H_2_O_2_-inducing reactive oxygen species (ROS)-mediated senescence was treated in MC3T3-E1 cells for 24 h. The formation of γH2AX foci, a double-stranded DNA marker, was confirmed using immunofluorescence (IF). Confocal microscopy also showed that the H_2_O_2_-treated cells qualitatively increased γH2AX foci formation compared with undamaged (untreated) cells (Figure 2B). After the analysis, the resulting ‘CSV’ files were displayed in the designated output path. The following data were derived from the CSV files: (1) out_green, (2) out_Min, and (3) out_Max. Because the analysis in Figure 2 was performed using single-wavelength images, the results for “out_red” were not used.

To quantitatively measure the accuracy of the γH2AX foci, over 100 cells were captured and compared using manual counting and ImageJ 1.53 software (Figure 3A,B). Using the Foci-Xpress, manual counting, and ImageJ methods, the ‘Control’ condition showed percentages of 4.5%, 2.65%, and 5.4%, respectively. For the ‘H_2_O_2_’ condition, the percentages were 73.5%, 69.4%, and 70.3%, respectively. Furthermore, there was no significant difference in the γH2AX foci performance among the methods (Figure 3A). We utilized Foci-Xpress to represent the average number of foci per entire nucleus. This approach serves as a widely accepted standard in radiation biology experiments and demonstrates our ability to detect damaged cells following H_2_O_2_ treatment (Figure 3C). Both sets of data revealed an increase in γH2AX foci due to treatment with the DNA-damaging agent H_2_O_2_ (Figure 3A,C). The Foci-Xpress method exhibited a processing speed approximately three times faster than that of the other methods for measuring foci (Figure 3B). Collectively, these results suggest that Foci-Xpress had a foci counting performance similar to that of the other methods, and the time required for counting was short.

### 2.3. Comparison of Processing Time Required When Increasing the Number of Images

We compared the analysis time of Foci-Xpress with that of manual counting and ImageJ software for the same set of images. We captured 5, 10, 20, 50, and 100 images and compared the analysis times for each method. The results demonstrated that the Foci-Xpress software could analyze the images approximately 2–3.8 times faster than manual counting and 3.4–6.3 times faster than ImageJ (Figure 4). These findings suggest that Foci-Xpress is significantly more efficient than traditional manual counting and ImageJ software for foci counting. A comparison of the times required for processing an increasing number of images suggest that there is a significant difference in processing time. In particular, when processing 50 images, Foci-Xpress takes approximately 22 min, whereas manual counting takes approximately 81 min and ImageJ takes 78 min. Thus, Foci-Xpress reduces the processing time and minimizes the time that researchers need spend on analysis. Foci-Xpress completes the task in less than 5 min. In contrast, manual counting and ImageJ analysis methods require researchers to dedicate the same amount of time as the processing time mentioned earlier, as they must be physically present during the analysis. These results suggest that Foci-Xpress is advantageous because of its automatic features.

### 2.4. The Performance of Telomere and γH2AX Colocalization after H_2_O_2_ Processing Can Be Measured Accurately and Promptly

In addition to γH2AX, proteins such as 53BP1, RAD50, and phospho-ATM form foci in the nuclei of a damaged cell [1,14,23,24,25,26]. Additionally, TIF, which causes telomere problems, is formed. The degree to which the cells’ telomeres were damaged was determined by the colocalization between γH2AX and the telomere. The telomeres were detected using in situ hybridization (ISH), and γH2AX was detected using IF. A TIF analysis defined a cell as positive if the intensity exceeded 10,000 with three or more foci in the nucleus (Figure 5A). The TIF analysis suggested that the damaged cells exhibited colocalization between γH2AX and their telomeres (Figure 5B). Along with the previous results, we analyzed the number of damaged cells or co-localizing telomere/γH2AX foci per cell in both untreated and damaged MC3T3-E1 cells after exposure to 50 µM of H_2_O_2_ for 24 h (Figure 6A,B). The results indicated that Foci-Xpress accurately and efficiently measured both the extent of γH2AX localization and colocalization of telomeres and γH2AX in response to H_2_O_2_ treatment. After the analysis, the ‘CSV’ files were reflected concurrently in the specified output path. The data were converted into four CSV files with the following file names: (1) out_green (result for γH2AX), (2) out_red (result for telomere probe), (3) out_Min, and (4) out_Max.

## 3. Discussion

We developed Foci-Xpress as a powerful tool in the field of molecular cell biology for quantifying nuclear foci within cellular environments. While other bundles have been developed for foci counting within imaging instruments, they have limitations, as they cannot be used independently and require an access fee. In contrast, Foci-Xpress is freely available to all users, making it a more accessible and valuable resource for researchers studying complex biological phenomena, such as cellular aging and cancer. Foci in nuclei are essential indicators of the cellular response to DNA damage and provide valuable insights into the underlying molecular mechanisms and various biological phenomena. For analysis purposes, we employed the protein γH2AX, which is widely recognized as a robust biomarker for assessing DNA damage and also forms foci within the nucleus [24]. In this study, we specifically focused on evaluating the speed, accuracy, and convenience of using Foci-Xpress. This tool demonstrated fast and precise results while saving considerable time and research effort.

The traditional method for quantifying damaged cells involves direct, manual counting. However, this method is prone to research bias. Various ImageJ-based solutions have been proposed to address these limitations; however, there are issues, such as infrequent updates and inaccurate results [27,28,29,30]. Foci-Xpress was developed to overcome the limitations of conventional analyzing tools. In this study, Foci-Xpress was utilized for an independent analysis of γH2AX and a colocalization analysis of γH2AX with telomeres. However, it is anticipated that Foci-Xpress will also be valuable for analyzing foci-forming proteins such as 53BP1, RAD50, and phosphorylated ATM in damaged cells, as well as for counting foci formed by proteins like RNA Pol II, promyelocytic leukemia protein (PML), fibrillarin, and proliferating cell nuclear antigen (PCNA) in undamaged cells [31,32,33,34]. The versatility of Foci-Xpress makes it a valuable tool for studying various aspects of foci formation in both damaged and undamaged cellular environments. These results suggest that it can also be useful for counting the foci of DNA damage response proteins, such as 53BP1, RAD50, and phospho-ATM, derived from damaged cells [1,5,14,26−27]. Genetic or pharmacological methods for cellular stress recovery induce a reduction in γH2AX foci. Moreover, Foci-Xpress quantitatively captured these changes in γH2AX foci that reflect cellular stress recovery [35,36].

In particular, the ability of Foci-Xpress to perform a colocalization analysis with other foci-forming probes offers valuable insights into the biological significance of DNA damage induction. By analyzing the spatial relationship between different foci formed within the nucleus, researchers can better comprehend the interconnections between various cellular processes and their contribution to the formation of nuclear foci. Foci-Xpress can uncover new molecular mechanisms that govern these processes and identify novel targets for therapeutic intervention.

An analysis of damaged cells based on the number of photos using Foci-Xpress demonstrated superior processing speed and accuracy compared to classical methods. We have confirmed that Foci-Xpress can implement both the method for quantifying the number of foci per cell and the method for quantifying the nuclei with ten or more γH2AX foci exceeding the specified intensity threshold. The former is widely recognized in radiation biology experiments, reaffirming our ability to detect damaged cells following H_2_O_2_ treatment. Increasing the number of images did not significantly affect the processing speed or accuracy of the analysis using Foci-Xpress for damaged cells. Moreover, the automated Foci-Xpress analysis allows researchers to streamline their analysis by determining the most critical parameters. This contrasts with classical methods, which can be time-consuming and error-prone when analyzing substantial amounts of data. This not only simplifies the analysis process but also enhances the accuracy and reliability of the obtained results. Using Foci-Xpress, researchers can easily analyze numerous images with a significant reduction in the time required to analyze the data in front of the analysis equipment. This demonstrates the convenience and efficiency of using an automated analysis program for research purposes. The ability to process large amounts of data promptly and accurately is particularly beneficial in fields where the analysis of numerous samples is required, such as drug development and disease diagnosis.

In this study, we developed an automated foci-counting tool using ImageJ software. Manual counting methods involve labor-intensive processing steps, in which researchers are required to manually adjust the parameters for each image; a time-consuming process that can introduce bias and inconsistencies due to repetitive procedural adjustments aimed at achieving statistical significance. To address these limitations, we focused on essential parameters such as minimum nuclear size, foci intensity, the number of foci within a nucleus, and the number of damaged cells. Our approach streamlines the parameter selection process, prioritizing the analysis of damaged cells while maintaining the speed and accuracy of existing methods. In our analysis pipeline, we leverage the robust ‘Particle Analysis’ feature of ImageJ, which is capable of identifying and categorizing particles of various sizes, ensuring a comprehensive evaluation of the data. It is noteworthy that our current program may define foci clusters as single foci. However, we anticipate that foci clustering is unlikely to significantly impact our research outcomes. Our criteria are specifically designed to identify damaged cells when there are 10 or more foci within a nucleus, exceeding a predefined intensity threshold. Although our program recognizes foci clusters as single entities due to the inherent limitations in ImageJ’s built-in functions, our counting process offers flexibility in adjusting parameters, including foci intensity, to classify foci based on their unique characteristics and perform multi-stage counting. Moreover, the analysis of colocalization between foci in multiple channels holds significance for gaining insights into additional biological processes. Furthermore, to overcome these limitations, we are actively pursuing the development of an enhanced program as a future research goal.

Foci-Xpress is a user-friendly interface, and its automated functions further facilitate the use of foci analysis programs, making it accessible to researchers without extensive computational experience. Overall, the ability of Foci-Xpress to simplify and automate the analysis process by allowing users to select only essential parameters underscores its value as a powerful tool for promptly and accurately analyzing substantial amounts of data.

## 4. Materials and Methods

### 4.1. Cell Culture and Treatment

A MC3T3-E1 cell line was obtained from ATCC (Manassas, VA, catalog #CRL-2593). The MC3T3-E1 cells were cultured with α-minimal essential medium (α-MEM) containing 10% (*v*/*v*) fetal bovine serum (FBS) and 1% (*v*/*v*) penicillin/streptomycin. The MC3T3-E1 cells were seeded on a cover glass at 2 × 10^4^ cell/cm^2^ in six-well plates. The following day, when the confluence of the cells was approximately 70–80%, the culture media was treated with 50 μM of H_2_O_2_ for 24 h (St. Louis, MO, USA, catalog #H1009). After 24 h, the cells were fixed with 4% paraformaldehyde. The fixed cells were subjected to IF.

### 4.2. Immunofluorescence Staining of γH2AX and Telomere Dysfunction-Induced Foci

For IF staining of γH2AX, MC3T3-E1 cells were cultured on coverslips, fixed for 15 min with 4% paraformaldehyde, and permeabilized for 15 min with 0.1% Triton X-100 in phosphate-buffered saline containing Tween 20 (PBST, 0.1% Tween 20). The cells were blocked with 3% bovine serum albumin (BSA) and incubated with Anti-γH2AX (Ser139) at a final concentration of 1:200 (cell signaling technology, catalog #9718). The secondary antibody reaction was performed using an Alexa Fluor 488 antibody (Invitrogen, Carlsbad, California, USA) in the dark. Subsequently, the nucleus was stained with DAPI and mounted.

Following the procedure mentioned earlier, the damaged cells underwent a secondary antibody assay to detect TIF. After the second antibody reaction, additional fixation was performed using 4% paraformaldehyde, and the cells were hybridized using a Cy3-labeled CCCTAA telomeric peptide nucleic acids (PNA) probe (Panagene, Daejeon, South Korea). To identify the nuclei, 4′,6-diamidino-2-phenylindole (DAPI) staining (ImmunoBioScience, Mukilteo, Washington, USA) was used. The γH2AX foci were quantified and visualized using a confocal microscope (LSM 800; Carl Zeiss, Oberkochen, Baden-Württemberg, Germany). All of the confocal microscopy images were captured at a magnification of 40×.

### 4.3. Quantification of Foci

After IF, the number of foci in the nuclei of a minimum of 100 cells was determined. After capturing an image using a confocal microscope, Foci-Xpress was used and compared with ImageJ and manual counting. In this study, we focused on analyzing foci using a 2D imaging approach and did not employ z-stack images for our analysis. Our software was specifically designed to perform foci analysis on images captured in a two-dimensional plane. To ensure the accuracy of foci quantification and to mitigate the possibility of double counting, we adopted a strategic imaging approach. Multiple images were captured from distinct non-overlapping regions, thereby minimizing the likelihood of foci being counted more than once. This approach enabled us to conduct foci analysis based on images taken from different sections, reducing the potential for any redundancy in foci quantification.

Classical methods using ImageJ and manual counting

Two methods were used to count the foci. First, manual counting of the image files was performed. In the manual method, damaged cells were manually defined, as in the case of more than 10 foci in the nucleus.

ImageJ software was used also for counting. In this method, the nucleus was selected by designating the ROI as DAPI. The nucleus selected within ROI was measured to the count of total cells, and the foci were captured using the command ‘Find maxima…’ tool. ImageJ-based quantification defined cells with 10 or more foci with an intensity of 10,000 or higher as damaged cells using ‘Find-maxima...’.

2.Foci-Xpress

In the Fiji-based counting tool, the sizes of the nuclei were measured, except for the pixel sizes within 100 µm^2^ based on the ImageJ pixels. For foci counting, cells with 10 or more foci and a degree of brightness of 10,000 or more were defined as damaged cells (Figure 1B). The parameters were defined for all of the confocal image files. After the parameters were set, Foci-Xpress separated the channels of the image (Figure 1A; step of the split channel). Subsequently, the nuclei stained with DAPI in each channel were defined as ROIs, as set in the parameters (Figure 1A; step of Detect Nucleus Area). The number of foci containing the marker (γH2AX) in the designated ROI was quantified (Figure 1A; steps for detecting the marker and acquiring marker data) using a preset parameter (intensity ≥10,000, 10 ≥ γH2AX foci; Figure 1B). The quantified data are presented as a single CSV file. All the parameters were applied to the confocal images for each independent experiment.

### 4.4. Development of a Novel Foci Quantification Tool

This script is based on ImageJ/Fiji software (National Institutes of Health) [37]. To read the image files, the BFImport function was used to import data from numerous life science file formats. In the image data, two or three channels represent the marker and background. In almost all cases, the blue channel indicates the background, and the red or green channel indicates the marker. To split and detect the channels, the getLut function in the ImageJ/Fiji script was used. The threshold of the Huang2 method [38], which is an alternative to the Huang method with changes for 16 bits, was used as the threshold to obtain temporary background binary images. Subsequently, the filling of holes and a Gaussian blur filter with a parameter of three sigmas were applied to reduce salt-and-pepper and random noises. The threshold of the Huang2 method was applied once more to obtain a background binary image with noise reduction. The watershed algorithm was applied to a noise-reduced background image to split the overlapping background [39]. Finally, the background was detected by analyzing particle modules in ImageJ with foci sizes that were predefined by users to count and measure objects from preprocessed binary or threshold images.

Marker images based on the ROIs of the detected background were saved to each image for subsequent processing. The marker images for each ROI were loaded separately. To process the marker images, Bernsen’s auto local threshold with a 40-pixel radius was applied to the marker image. The markers were detected using particle analysis modules in ImageJ to count and measure objects from predefined binary or threshold images. Because the marker particles were smaller, there was no size limitation. If both red and green marker images were available, the minimum and maximum values of the two marker intensities were obtained.

Each experimental image was processed separately. After the separation process, each experiment was summarized using the marker property data. Mark brightness and counts were used for summarization and selection. The markers were selected based on criteria for marker brightness. Finally, the program made a recapitulation comprising the average intensity, the number of selected nuclei, the number of total nuclei, and the ratio of the selected nuclei.

## Figures and Tables

**Figure 1 ijms-24-14465-f001:**
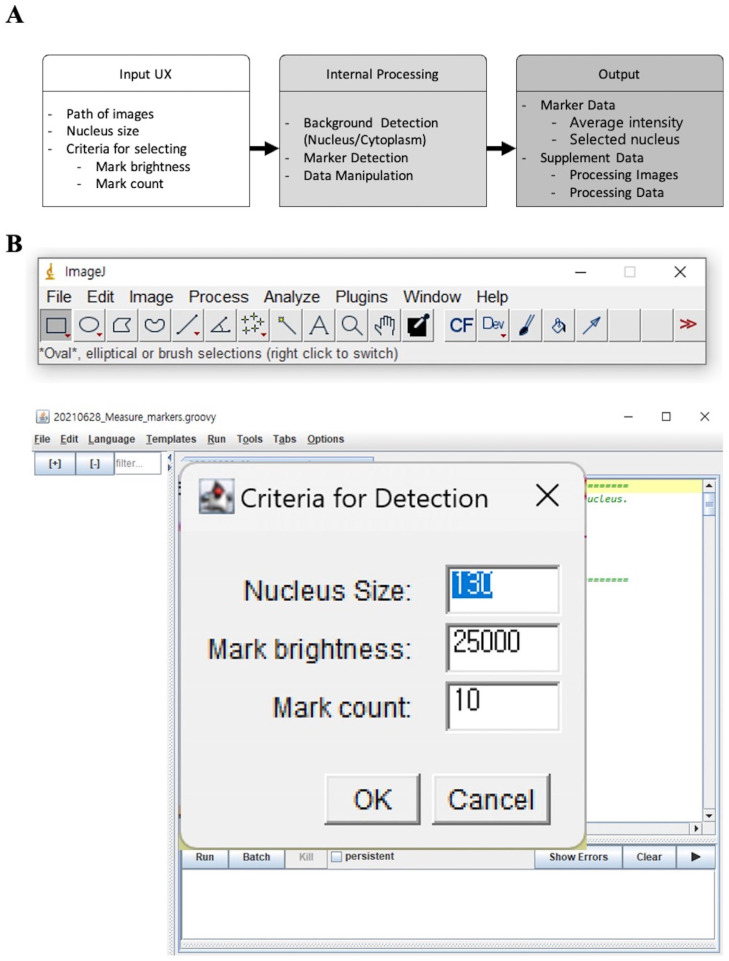
Overview of the software system: (**A**) workflow of Foci-Xpress; (**B**) main interface of Foci-Xpress.

**Figure 2 ijms-24-14465-f002:**
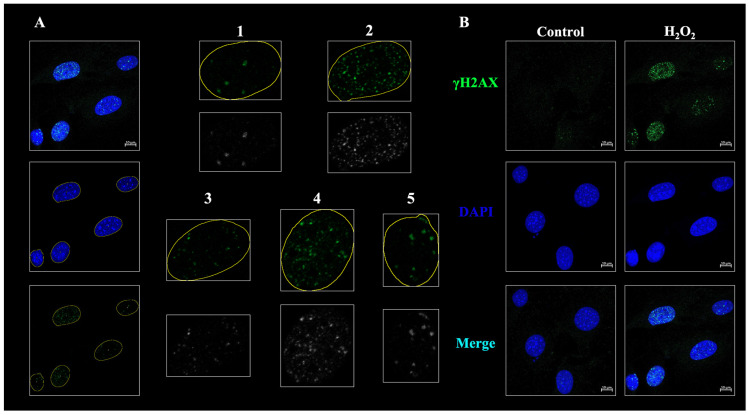
Representative images of the working process of Foci-Xpress for capturing the nucleus in confocal microscopy images. (**A**) Image capture process for foci counting. The image represented merges DAPI and γH2AX. Files used by CZI estimated the regions of interest (ROI) value for the nuclei in the yellow border. Each nucleus is numbered to determine the number of nuclei. For foci counting, the channel of γH2AX images was converted into greyscale images. γH2AX foci in the nucleus were estimated based on the following parameters: nucleus size, foci intensity, and the number of foci in nuclei. Scale bar: 10 μm. Immunofluorescence was performed using a γH2AX foci (grren) and DAPI (blue). (**B**) Representative images of immunofluorescence (IF) performed on M3CTC-E1 cells. MC3T3-E1 cells were treated with 50 µM of H_2_O_2_ for 24 h. MC3T3-E1 cells stain γH2AX (phosphorylation of H2AX at ser139) in green (top image) and DAPI in blue (middle image). The image of the merged channels is shown at the bottom. Scale bar: 10 μm.

**Figure 3 ijms-24-14465-f003:**
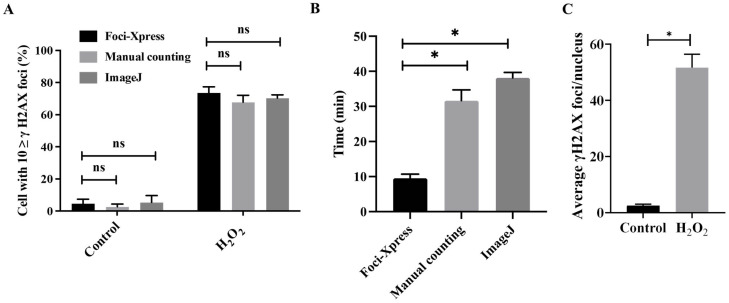
Analysis of damaged cells compared with classical methods. (**A**) Graph of the foci formation of γH2AX in cells treated with or without H_2_O_2_, as shown in Figure 2B. Error bars show the SEM of two duplicates, and ‘n.s’ indicates no significance. (**B**) Quantification of measurement time between the different counting methods. (**C**) The average number of γH2AX foci per cells for 100 cells treated with or without H_2_O_2_, as shown in Figure 2B. Error bars show the SEM from two duplicates and indicate a * *p*-value < 0.05.

**Figure 4 ijms-24-14465-f004:**
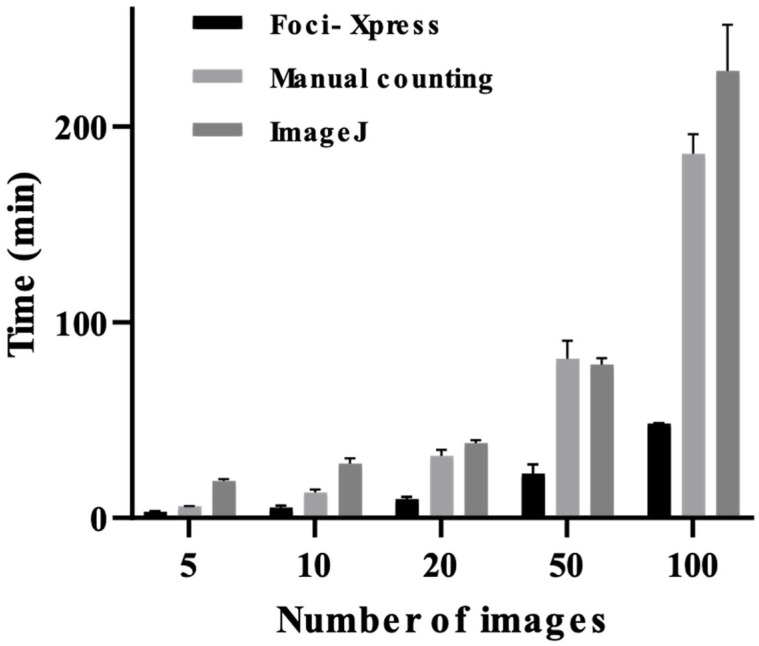
Analysis of the time required for Foci-Xpress. A minimum of 10 images per group were randomly captured using a 40× objective lens following IF staining. The samples were imaged using a confocal microscope (LSM 800, Carl Zeiss). The same set of images was analyzed using three independent methods for each group: 5, 10, 20, 50, and 100 images.

**Figure 5 ijms-24-14465-f005:**
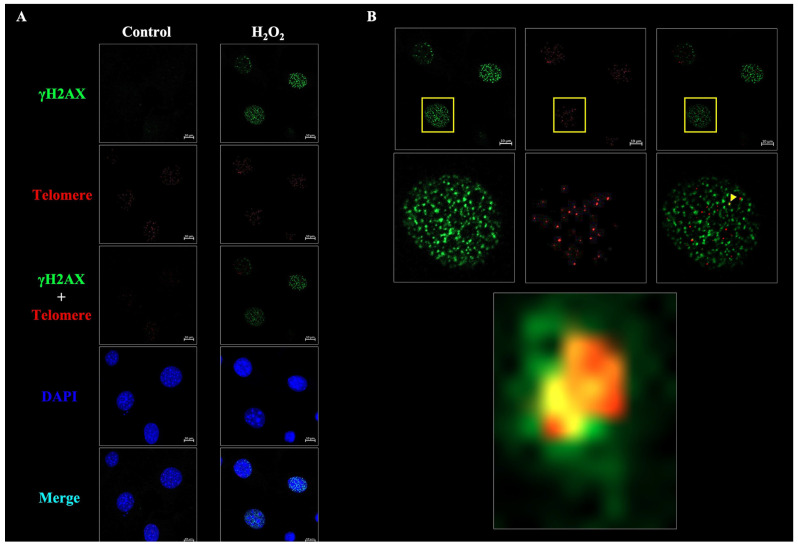
Representative confocal microscopy images of TIF. (**A**) Representative colocalization images of TIF. MC3T3-E1 cells were treated with 50 µM of H_2_O_2_ for 24 h or were untreated. MC3T3-E1 cells were incubated with anti-γH2AX (phosphorylation of H2AX at ser139) in green (top image) and DAPI in blue (4th image from the top). Images of the middle and bottom are merged channels green + red or green + red + blue images. Scale bar: 10 μm. Immunofluorescence was performed using a γH2AX foci (grren), telemore probe (red) and DAPI (blue). (**B**) Representative colocalization of damaged cells with labeled γH2AX foci (green) and telomere DNA (red). Scale bar: 10 μm.

**Figure 6 ijms-24-14465-f006:**
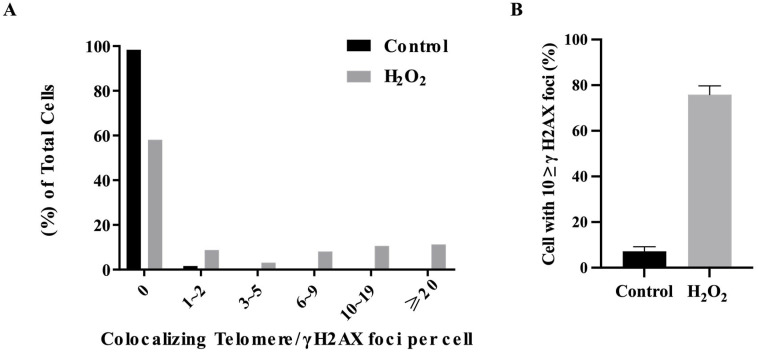
Graphs of colocalized foci and foci counting analysis at a single wavelength using Foci-Xpress. (**A**) Percentage of colocalization between γH2AX and the telomeric probe in the cells captured in Figure 5. The grey bars represent the H_2_O_2_ group, whereas the black bars represent the untreated control group. H_2_O_2_: cells treated with 50 µM of H_2_O_2_ for 24 h. (**B**) Percentage of γH2AX in the cells captured in Figure 5. The grey bars represent the H_2_O_2_ group, whereas the black bars represent the untreated control group. H_2_O_2_: cells treated with 50 µM of H_2_O_2_ for 24 h.

## Data Availability

The information regarding Foci-Xpress is accessible to the public at: “https://github.com/Jae-I/ERASMUS”. It will be accessible from 19 June 2023.

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
