# Peer review of "Foci-Xpress: Automated and Fast Nuclear Foci Counting Tool"

_ijms, 2023, doi:10.3390/ijms241914465_

Round 1

Reviewer 1 Report

Moon et al describe Foci-Xpress, open-source software they have developed that, when used in conjunction with the long-existing program, ImageJ, allows automated quantification of nuclear protein foci in multiple images.  They provide evidence that Foci-Xpress generates quantitative data on the number and intensity of nuclear foci in samples stained for gamma-H2AX equivalent to that derived by manual counting, or individual image analysis with ImageJ, but significantly decreases the time needed to perform the analysis.  They also argue that automated counting eliminates the risk of human bias during manual counting.  Subsequent expts show that Foci-express can also quantify co-localizing signals after DNA damage using gamma-H2AX and telomere RNA hybridization as examples.

In general, the experiments have been carefully performed and interpreted appropriately.  The discussion is rather repetitive, basically stating that “quantifying nuclear foci can provide important insight into various biological processes” multiple times – some abbreviation and streamlining would improve the aesthetic of the paper.  Similarly, although the English language usage is generally good, there are a few exceptions where the scientific meaning is unclear.  Two examples:

1) lines 143-145 “Under un-damaged conditions, approximately 4.5% of the cells had over 10 γH2AX foci, and approximately 73.5% of the cells were damaged” this doesn’t really make sense compared with the data.

2) lines 249-251 “According to our previous report [36], the recovery effect based on the number of γH2AX foci could be confirmed by γH2AX formation by Padi2 and RelA siRNA, and by the formation of γH2AX foci reduced by chemical (Bay) treatment. Therefore, the performance of Foci-Xpress is verified in a study using a senescence model induced by H2O2 in MC3T3-E1 cells” very unclear what is meant here – possibly reading reference 36 might help to figure it out, but it’s bad practice to include statements in a manuscript that cannot be understood without reference to external information.

This is not an exhaustive list so my suggestion is the ms be edited by a native English speaker to eliminate these little quirks.

Overall the english language usage is pretty good, although there are a few instances where scientific meaning is obscured or cryptic - I have given two examples in "comments to authors", although this is not an exhaustive list.  My suggestion would be that a native english speaker edit the paper to eliminate these quirks.  It shouldn't be a very big task.

Author Response

1) lines 143-145 “Under un-damaged conditions, approximately 4.5% of the cells had over 10 γH2AX foci, and approximately 73.5% of the cells were damaged” this doesn’t really make sense compared with the data.

Answer : We appreciate your attention to the discrepancy that you've pointed out and we sincerely apologize for any confusion that may have arisen from the initial statement. The revised our description now accurately reflect the data acquired through the diverse quantification approaches.

In the specific context of lines 143-147, it is noted that “Using the Foci-Xpress, Manual Counting, and ImageJ methods, the 'Control' condition showed percentages of 4.5%, 2.65%, and 5.4%, respectively. For the 'H2O2' condition, the percentages were 73.5%, 69.4%, and 70.3%, respectively. Furthermore, there was no significant difference in the γH2AX foci performance among the methods”.

2) lines 249-251 “According to our previous report [36], the recovery effect based on the number of γH2AX foci could be confirmed by γH2AX formation by Padi2 and RelA siRNA, and by the formation of γH2AX foci reduced by chemical (Bay) treatment. Therefore, the performance of Foci-Xpress is verified in a study using a senescence model induced by H2O2 in MC3T3-E1 cells” very unclear what is meant here – possibly reading reference 36 might help to figure it out, but it’s bad practice to include statements in a manuscript that cannot be understood without reference to external information.

Answer :

We greatly appreciate your valuable comment and have taken diligent measures to improve the clarity and accessibility of this manuscript.

(Lines 249-251) In response to your feedback, we have revised the manuscript as follows: "Genetic or pharmacological methods for cellular stress recovery induce a reduction in γH2AX foci. Moreover, Foci-Xpress has quantitatively captured these changes in γH2AX foci that reflect cellular stress recovery [36,37].

Overall the english language usage is pretty good, although there are a few instances where scientific meaning is obscured or cryptic - I have given two examples in "comments to authors", although this is not an exhaustive list.  My suggestion would be that a native english speaker edit the paper to eliminate these quirks.  It shouldn't be a very big task.

Answer :

Thank you for pointing out these issues. We have received your feedback and appreciate it. We have utilized the services of 'Editage,' an English editing service staffed by native English speakers, to review our manuscript. You can access this information through the link shared on Google Drive. Thank you again for your valuable assistance.

https://drive.google.com/file/d/1fH86vB26S2H3MDj_J-D77Dq-nYPMGzAV/view?usp=sharing 

Reviewer 2 Report

The manuscript presents an automated foci-counting tool using ImageJ software. Foci-Xpress aims to enable the processing of multiple foci images in a single run using automated counting. It also allows the analysis of nuclear size and foci intensity and number with minimal parameter settings.

The model the authors used for validation is using telomere dysfunction-induced foci using H2O2 not radiation induced foci. Unsure how appropriate this is for the special edition considering the authors have not used radiation once. This is one of the limitations of  this study: DNA damage foci formed by  radiation are generally known to have different sizes and, considering different radiation qualities, the DNA damage foci have quite a wide range of sizes and clusters. These are the situations where issues using automated foci analysis arise. Considering different imaging planes and different foci sizes, the foci quantification automation is an ongoing challenge, and it would help immensely this to be addressed.

The manuscript is well written and presents an interesting software, needed in the radiation biology field. However, the authors need to acknowledge the limitations of the model they used and aim to improve and address the entire complexity of foci counting.

A few specific comments below:

1.       The percentage of cells having over a certain number of foci is a ‘lazy’ approach, and manually isn’t as time consuming as quantifying the foci number per cell.  For an automated foci counting, the authors do not present any foci quantification in terms of foci per cell for 50 or 100 cells, which is the standard for a radiation biology experiment.

2.       Have the authors used z-stack images for the foci analysis presented here? Is the software going to be able to accurately count different planes of the same cell? Is this going to lead to double counting some of the foci?

3.       In Figure 3, for the percentage of cells with over 10 foci, how many cells and images were initially on the images? How many images?

4.       What is the magnification necessary for the acquired images to enable a correct quantification?

5.       What would happen in a model where the foci cluster? Or there are different foci sizes throughout the same nucleus?

English usage is appropriate, however, minor edits for typos etc need to be addressed.

Author Response

1. The percentage of cells having over a certain number of foci is a ‘lazy’ approach, and manually isn’t as time consuming as quantifying the foci number per cell.  For an automated foci counting, the authors do not present any foci quantification in terms of foci per cell for 50 or 100 cells, which is the standard for a radiation biology experiment.

Answer :

I appreciate your feedback and would like to clarify our approach. It appears that there might have been some misunderstanding.

In the context of radiation biology experiments, your description of foci counting strategies seems to pertain to situations such as examining the dynamics of time-dependent or dose-dependent responses, which has been commonly utilized in previous studies. In those cases, analyzing patterns of foci reduction over time could indeed provide meaningful insights. However, our experimental design also aligns with a foci counting approach commonly utilized in radiation biology experiments [1-3].

In our study, we counted the number of damaged cells, which we defined as cells with a certain number of foci. To achieve this, we selected the number of foci per nucleus as a parameter. If a cell had more than a specified threshold of foci, we categorized it as a damaged cell during our counting process. We should note that we did not set multiple time points or intensities, as our primary goal was to distinguish between undamaged and damaged cells. Therefore, we defined a nucleus as 'damaged' when it contained a specific number of foci or more. This criterion was established to serve the purpose of differentiating between these two conditions within a nucleus.

Altogether, as I mentioned earlier, quantifying the total number of foci was not our intended criterion. Counting the total number of foci across all cells is not meaningful for our specific research objectives. For example, within a set of 10 nuclei, one nucleus exhibits 50 foci, while the remaining five nuclei each contain 10 foci. In this scenario, when calculating the total number of foci relative to the number of cells, both conditions may be perceived as damaged cells. However, it's important to note that the former case does not align with our concept of damaged cells within the scope of our research.

We genuinely appreciate your valuable insights and feedback. Your thoughtful contributions greatly contribute to our understanding of the research. If you have any further inquiries or suggestions, please do not hesitate to reach out. Your engagement is invaluable to us.

2. Have the authors used z-stack images for the foci analysis presented here? Is the software going to be able to accurately count different planes of the same cell? Is this going to lead to double counting some of the foci?

Answer : We appreciate your inquiry regarding the use of z-stack images for our foci analysis.

In our study, we did not employ z-stack images for this purpose. Our experimental design involved seeding cells in a monolayer, where the majority of nuclei were situated within the same plane. Consequently, during the microscopy imaging process, we carefully selected the focal plane that provided the most optimal visualization of nuclei. This selection ensured that our images were captured in a 2D imaging plane, where nuclei appeared uniformly and prominently.

Regarding the accurate counting of foci across different planes of the same cell, we took precautions to prevent any potential double counting. We strategically captured images from distinct regions to ensure that foci were not measured redundantly. By adopting this approach, we aimed to minimize the possibility of foci being counted more than once.

In response to your feedback, we have made significant revisions to the manuscript. A comprehensive description of our methodology, addressing these considerations, has been added to the 'Quantification of Foci' section within the Materials and Methods (lines 328-336). We believe that these changes enhance the clarity and rigor of our study's methodology.

We have revised the manuscript as follows: In this study, we focused on analyzing foci using a 2D imaging approach and did not employ z-stack images for our analysis. Our software was specifically designed to perform focus analysis on images captured in a two-dimensional plane. To ensure the accuracy of foci quantification and to mitigate the possibility of double counting, we adopted a strategic imaging approach. Multiple images were captured from distinct non-overlapping regions, thereby minimizing the likelihood of foci being counted more than once. This approach enabled us to conduct focus analysis based on images taken from different sections, reducing the potential for any redundancy in foci quantification.

3. In Figure 3, for the percentage of cells with over 10 foci, how many cells and images were initially on the images? How many images?

Answer: We have taken your valuable feedback into consideration and made necessary adjustments to the manuscript.

In response to your question about Figure 3, in the 'Control' condition, our analysis was conducted on a total of 111 cells from 24 images. For the 'H2O2' condition, we analyzed a dataset comprising 121 cells from 33 images.

It's important to note that the ' H2O2' condition, characterized by oxidative stress induction, can create a toxic environment for cells, potentially leading to cell death. As a result, a smaller number of viable cells were available for imaging in the ' H2O2' condition compared to the control condition. To ensure the statistical robustness of our analysis and to obtain representative results, we decided to capture a greater number of images in the ' H2O2' condition. This approach allowed us to account for the reduced availability of cells while still maintaining a meaningful sample size for analysis.

We appreciate your attention to these details and your interest in understanding the dataset. If you have any further inquiries or require additional information, please do not hesitate to reach out. Your thoughtful questions contribute to a deeper understanding of our research.

4. What is the magnification necessary for the acquired images to enable a correct quantification?

Answer: We greatly appreciate your discerning feedback, which has significantly contributed to the enhancement of our study.

As reviewer mentioned, we have revised the "Materials and Methods" section accordingly.

(Lines 323-324) In our study, a magnification of 40X was employed, strategically chosen to strike a harmonious balance between resolution and field of view. This judicious magnification choice facilitated the distinct visualization of both foci and cellular structures. Consequently, we achieved an accurate identification and meticulous quantification of the observed features.

Specifically, this information has been included in the "Materials and Methods" section, under the subheading "4.2. Immunofluorescence staining of γH2AX and telomere dysfunction-induced foci." We sincerely thank you for your insightful input, which has led to the refinement of our methodology and enriched the scientific integrity of our work.

5. What would happen in a model where the foci cluster? Or there are different foci sizes throughout the same nucleus?

Answer :

(Lines 273-293) We have incorporated these modifications in the fifth paragraph of the “Discussion” section.

We sincerely appreciate your astute observations and questions, which have prompted us to provide further clarification. Moreover, your query about different foci sizes within the same nucleus is indeed crucial. In our analysis pipeline, we utilize the 'Analyze Particles' function in ImageJ, a powerful tool that enables us to identify and categorize particles of various sizes. This function ensures that all sizes of foci are detected and treated as distinct particles, thus contributing to a comprehensive assessment of the data.

In the case of foci clustering, it should be noted that when multiple foci aggregate, they are considered as a single entity in our analysis. However, we believe that foci clustering is unlikely to significantly impact the outcomes of our study. Our counting criteria are designed to identify damaged cells when there are 10 or more foci within a nucleus, surpassing a certain intensity threshold.

It's worth noting that while our current program recognizes foci clusters as single entities, we acknowledge the need for more sensitive counting in certain scenarios. If the research demands it, we are open to updating our methods to accommodate these requirements.

In addition, the situation where foci are counted as a single entity is attributed to the limitations of the ImageJ software's built-in functions. Nevertheless, our counting process allows for the adjustment of parameters, including foci intensity, which enables us to categorize foci based on their characteristics and conduct counting in multiple stages.

Thank you for your valuable input and questions. Your feedback greatly contributes to the rigor of our study, and we appreciate the opportunity to address these points of clarification.

English usage is appropriate, however, minor edits for typos etc need to be addressed.

Answer :

Thank you for pointing out these issues. We have received your feedback and appreciate it. We have utilized the services of 'Editage,' an English editing service staffed by native English speakers, to review our manuscript. You can access this information through the link shared on Google Drive. Thank you again for your valuable assistance.

https://drive.google.com/file/d/1fH86vB26S2H3MDj_J-D77Dq-nYPMGzAV/view?usp=sharing 

  1. Xue, X., et al. Geomagnetic Shielding Enhances Radiation Resistance by Promoting DNA Repair Process in Human Bronchial Epithelial Cells. International Journal of Molecular Sciences, 2020. 21, DOI: 10.3390/ijms21239304.
  2. Rai, P., et al., <i>Streptococcus pneumoniae</i> secretes hydrogen peroxide leading to DNA damage and apoptosis in lung cells. Proceedings of the National Academy of Sciences, 2015. 112(26): p. E3421-E3430.
  3. Iriki, T., et al., Senescent cells form nuclear foci that contain the 26S proteasome. Cell Reports, 2023. 42(8): p. 112880.

Round 2

Reviewer 2 Report

The authors have amended their manuscript, aiming to improve the clarity, however the changes are not tracked on the second version of the manuscript, and therefore are not clear. After reading the authors rebuttal, the objective of their research becomes very unclear.

Firstly, the authors state, in their rebuttal that ‘quantifying the total number of foci was not our intended criterion. Counting the total number of foci across all cells is not meaningful for our specific research objectives.’ . All this while the title currently is ’Foci-Xpress: Automated and fast nuclear foci counting tool.’

Furthermore, the authors have submitted the manuscript to a radiation and DNA damage special edition, where the standard is foci quantification and repair kinetics. The authors repeat the words ‘Quantification of foci’ while they clearly state this is not their aim. Maybe use the word analysis, and not have a title implying foci counting, while this is not what their tool does.

It is not clear what this manuscript brings to the field. Every lab has their own protocol to quantify DNA damage, from the staining protocol to the quantification, which are always more or less automated.

The authors discuss differential DNA damage levels, their quantification, in a way that suggests limited knowledge to this very simple aspect of radiation biology. Furthermore, using H2O2 as a substitute for radiation is not a very convincing model, especially for a radiation based special edition.

Minor english nuances, such s using the word 'counting' while the authros state that is not what their tool does really hamper the clarity of the manuscript.

Author Response

Before Reviewer’s Comment: We apologize for our lack of understanding in addressing the reviewer's comments initially. In response to the reviewer's suggestion to include "foci quantification in terms of foci per cell for 50 or 100 cells" in the first review, we have added a new Figure 3C.

[Figure 3C and Line 170-171] A newly added figure representing the average number of foci counted across 100 cells, along with an explanation for it.

(C) The average number of γH2AX foci per cells for 100 cells treated with or without H2O2 as a result of Figure 2B

Reviewer Comment 1: Firstly, the authors state, in their rebuttal that ‘quantifying the total number of foci was not our intended criterion. Counting the total number of foci across all cells is not meaningful for our specific research objectives.’. All this while the title currently is ’Foci-Xpress: Automated and fast nuclear foci counting tool.’ Furthermore, the authors have submitted the manuscript to a radiation and DNA damage special edition, where the standard is foci quantification and repair kinetics. The authors repeat the words ‘Quantification of foci’ while they clearly state this is not their aim. Maybe use the word analysis, and not have a title implying foci counting, while this is not what their tool does.

Answer: It appears there may have been a misinterpretation of our previous statement on the part of the reviewer, 'quantifying the total number of foci was not our intended criterion. Counting the total number of foci across all cells is not meaningful for our specific research objectives.'

As we explained following that sentence in our first response to the reviewer, our software is not intended to 'avoid counting' but rather, its basic function includes foci counting and extends to analyzing groups of damaged cells by reflecting the even distribution of the number of foci per cell across the entire sample. This approach has been adopted in previously published papers as well [1,2].

Our Foci-Xpress is versatile, offering reliable analytical results across various research perspectives. As shown in Figure 3, it also demonstrated a clear difference in damaged cells using the method suggested by the reviewer. Therefore, it is appropriate to describe our software as 'Foci-Xpress: Automated and fast nuclear foci counting tool.'

To reflect the reviewer’s concerns, we have added the following statement to the manuscript.

Line 147-152 and 171-172: We utilized the Foci-Xpress to represent the average number of foci per entire nucleus. This approach serves as a widely accepted standard in radiation biology experiments and demonstrates our ability to detect damaged cells following H2O2 treatment (Figure 3C).

Line 265-270: We have confirmed that Foci-Xpress can implement both the method for quantifying the number of foci per cell and the method for quantifying the nuclei with ten or more γH2AX foci exceeding the specified intensity threshold. The former is widely recognized in radiation biology experiments, reaffirming our ability to detect damaged cells following H2O2 treatment.

We hope this explanation adequately addresses your concerns.

1) Xue, Xunwen, et al. "Geomagnetic shielding enhances radiation resistance by promoting DNA repair process in human bronchial epithelial cells." International Journal of Molecular Sciences 21.23 (2020): 9304.

2) Rai, Prashant, et al. "Streptococcus pneumoniae secretes hydrogen peroxide leading to DNA damage and apoptosis in lung cells." Proceedings of the National Academy of Sciences 112.26 (2015): E3421-E3430.

Reviewer Comment 2: The authors discuss differential DNA damage levels, their quantification, in a way that suggests limited knowledge to this very simple aspect of radiation biology. Furthermore, using H2O2 as a substitute for radiation is not a very convincing model, especially for a radiation based special edition.

Answer: The decision to designate our research for the 'Special Topic: Radiation-Induced Cellular Damage, Repair and Responses' section was made by the editorial office. Prior to the invited submission, we consulted with the editor of IJMS (Dr. Demi Dong), and we respect their judgment that the aims and content of our manuscript are suitable for this Special Topic.

Furthermore, as reported from previous studies, some of the mechanisms of DNA damage induced by radiation involve the generation of ROS (Reactive Oxygen Species). Therefore, it cannot be said that the H2O2 treatment model, which represents DNA damage by a typical ROS, is entirely unrelated to this Special Topic.

1) “Ogawa, Yasuhiro. "Paradigm shift in radiation biology/radiation oncology—exploitation of the “H2O2 effect” for radiotherapy using low-LET (linear energy transfer) radiation such as X-rays and high-energy electrons." Cancers 8.3 (2016): 28. “

2) “Jia, Chengyou, et al. "The role of DNA damage induced by low/high dose ionizing radiation in cell carcinogenesis." Exploratory Research and Hypothesis in Medicine 6.4 (2021): 177-184. “

3) Kim, Wanyeon, et al. "Cellular stress responses in radiotherapy." Cells 8.9 (2019): 1105.

The paper also discusses the formation of H2O2 by radiation. In fact, the paper cites "Zou, Z.; Chang, H.; Li, H.; Wang, S. Induction of reactive oxygen species: An emerging approach for cancer therapy. Apoptosis 2017, 22, 1321–1335" as an example of a study that supports this claim.